# THEORETICAL FOUNDATIONS OF CURRICULUM LEARNING IN LINEAR RNNS

## ABSTRACT

Pretraining models with a curriculum of simpler tasks is a common approach to speed up training. However, it is unclear what aspects of task structure drive learning speed, and how to practically choose the curriculum based on theoretical principles. Using recent advances in the analysis of learning trajectories in linear RNNs (Proca et al., 2025), we study a simple but informative example of performing two integration tasks in sequence, and ask what aspects of their task structure lead to faster overall learning of the second "target" task. We show both analytically and through simulations that even for tasks that are similar in their geometry, sequencing them based on the strength and scale of the input-to-target correlations can provably enhance learning speed. A surprising result from our theory that goes against conventional wisdom is that training intermediate tasks to suboptimal accuracies can be more beneficial to learning speed, rather than training them to convergence. These results provide foundational insight into how task similarity forms both a theoretical and practical basis for curriculum learning.

## 1 INTRODUCTION

Efficiently training neural network models on complex tasks can be difficult. One approach that often proves useful in practice is pretraining on simpler related tasks. Curriculum learning (CL), or pretraining more generally, are now ubiquitously used across many domains in machine learning (Soviany et al., 2022; Hacohen & Weinshall, 2019; Narvekar & Stone, 2018). Yet, documented counterexamples show that CL does not always help Wu et al. (2020). What makes for good pretraining tasks and how to construct effective curricula remain an open area of study, not only in machine learning but also in cognitive and neural science (Ferguson, 1956; Dekker et al., 2022; Behrens et al., 2018; Kepple et al., 2022).

One main reason for this limited understanding is that the effects of curriculum training are almost entirely assessed through simulations, which makes the extraction of general principles difficult. Some progress has been made recently in understanding how feedforward networks can see training speedups due to curricula (Lee et al., 2024; Saglietti et al., 2022), or though structured initial conditions(Liu et al., 2024); however a similar mathematical apparatus that can describe recurrent neural network (RNNs) learning has long been missing. Very recent advances in the analysis of learning dynamics for linear RNNs by Proca et al. (2025) open the door for starting to think about effects of RNN pretraining in precise mathematical terms.

Existing accounts of CL pretraining largely frame its success in terms of regularizing the loss landscape (Bengio et al., 2009): simpler pretaining tasks are assumed to have smoother loss surfaces in which solutions are easy to locate. This in turn provides favorable initial conditions for parameter optimization in the target task. While this description seems intuitive, it does assume that the loss landscapes (or at least the regions of good solutions) are well aligned across tasks. It is not clear how to assess this notion of task similarity outside of actually training the model on the two tasks. This brings up more general (and largely unanswered) questions about what makes a pretraining task similar to the target and is the alignment of the losses the only way to measure it?

In this work we build on analytical solutions for the learning dynamics of input and output parameters in linear RNNs to ask in precise mathematical terms how long does it take for a given task to train to convergence either directly or via an intermediate pretraining task (Figure 1A). In this framing, task similarity is naturally defined in terms of input and output covariances, which allows

for a general treatment of CL in this class of problems in terms of the geometry and alignment of these covariances across tasks.

Our approach is organized in the following way: First, we briefly summarize the problem of optimizing the input and output weights of RNNs. Next, we derive our core result that demonstrates how long it takes to optimize RNNs after they have already learned a separate task with related structure. We then detail the dimensions of task similarity that most drive fast learning. Finally, we explore the generalization of these training insights beyond the scope of our theory by studying training with nonlinear RNNs. In this work we contribute to the fundamental theoretical understanding of curriculum learning, highlight the significance of task similarity upon its success, and demonstrate practical principles for choosing the sequence of tasks for effective curricula.

## 2 CURRICULUM LEARNING DYNAMICS IN LINEAR RNNS

### 2.1 PROBLEM FORMULATION

Consider the dynamics of a linear RNN (Fig. 1B):

$$\boldsymbol{h}_t = \boldsymbol{W}_h \boldsymbol{h}_{t-1} + \boldsymbol{W}_x \boldsymbol{x}_t \tag{1}$$

$$\boldsymbol{y}_t = \boldsymbol{W}_y \boldsymbol{h}_t, \tag{2}$$

which maps time-varying inputs $\boldsymbol{x}_t \in \mathbb{R}^{N_x \times 1}$ into a network state $\boldsymbol{h}_t \in \mathbb{R}^{N_h \times 1}$, read out into outputs $\boldsymbol{y}_t \in \mathbb{R}^{N_y \times 1}$. The parameters of the network include the recurrent weight matrix $\boldsymbol{W}_h \in \mathbb{R}^{N_h \times N_h}$, input matrix $\boldsymbol{W}_x \in \mathbb{R}^{N_h \times N_x}$, and output matrix $\boldsymbol{W}_y \in \mathbb{R}^{N_y \times N_h}$.

We will focus on a family of tasks in which input streams $\boldsymbol{x}_{1:T}$ are integrated over time with different linear filters to yield outputs, $\hat{\boldsymbol{y}}_T$, at the end of the trial, $T$.[1] The loss over a batch of $P$ trials for this single output scenario of generating a target $\boldsymbol{y}$ is given as:

$$\mathcal{L} = \frac{1}{2} \sum_p^P \|\boldsymbol{y}_p - \hat{\boldsymbol{y}}_{T,p}\|^2. \tag{3}$$

Network parameters are optimized by backpropagation through time to minimize this objective.

**Covariances as fundamentals of task structure.** Starting from initial state $\boldsymbol{h}_0 = \boldsymbol{0}$, the network dynamics evolve as $\boldsymbol{h}_t = \sum_{i=1}^{t} \boldsymbol{W}_h^{t-i} \boldsymbol{W}_x \boldsymbol{x}_i$, which allows the loss to be rewritten as

$$\mathcal{L} = \sum_{t,t'=1}^{T} \frac{1}{2} \operatorname{Tr} \left[ \boldsymbol{W}_y \boldsymbol{W}_h^{T-t} \boldsymbol{W}_x \boldsymbol{\Sigma}_{x_t x_{t'}} \boldsymbol{W}_x^\top \boldsymbol{W}_h^{T-t'\top} \boldsymbol{W}_y^\top - \boldsymbol{W}_y \boldsymbol{W}_h^{T-t} \boldsymbol{W}_x \boldsymbol{\Sigma}_{x_t y} \right] + \text{const.} \tag{4}$$

This expression directly highlights what changes in the loss function from task to task: the covariances among inputs $\boldsymbol{\Sigma}_{x_t x_{t'}}$, and the input-output covariance $\boldsymbol{\Sigma}_{x_t, y}$.[2] Specifically, the input covariance function $\boldsymbol{\Sigma}_{x_t x_t'} = \mathbb{E}[\boldsymbol{x}_t \boldsymbol{x}_{t'}^\top]$ is an $N_x \times N_x \times T \times T$ tensor that captures how inputs co-vary with each other across both input channels, as well as time. The cross-correlation between time-varying input and the target output, $\boldsymbol{\Sigma}_{x_t y} = \mathbb{E}[\boldsymbol{x}_t \boldsymbol{y}_T^\top]$ is an $N_x \times N_y \times T$ tensor that captures how inputs, from each input channel and at every time point, relate to targets across different output channels.

Previous approaches assume white noise in the inputs by shifting any temporal dependence into $\boldsymbol{\Sigma}_{x_t y}$ (Proca et al., 2025; Saxe et al., 2014), but here we need to consider full spatial and temporal correlations in $\boldsymbol{\Sigma}_{x_t x_t'}$. This is an unavoidable consequence of our multi-task setup: while it is possible to rotate the coordinates to whiten input for a single task, it is not generally possible to find a single rotation will whiten them for both tasks. Since we are studying the learning dynamics of tasks in sequence, we must embrace the temporal dependence in the inputs.

The general goal of our derivation is to determine the time $\tau$ that it takes a network to learn a target task "2", and to contrast that learning time with a scenario where the network starts by training

---

[1] A generalization to continuous outputs is in principle possible, see Proca et al. (2025) Appendix M.

[2] Note that we will refer to the transpose of the matrix $\boldsymbol{\Sigma}_{x_t y}$ for a fixed time point via its indices as $\boldsymbol{\Sigma}_{x_t y}^\top = \boldsymbol{\Sigma}_{y x_t}$.

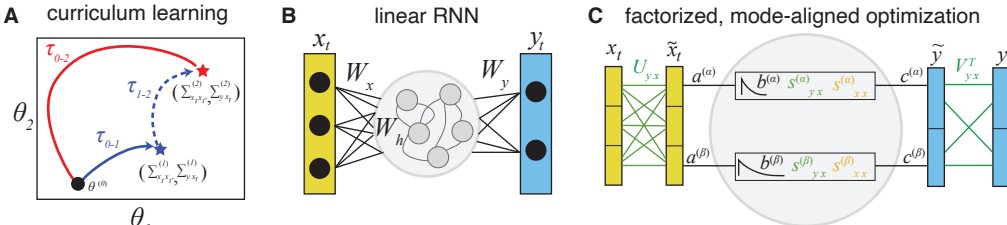

Figure 1: **A.** Curriculum learning from the lens of task similarity. The time $\tau$ needed to learn a task $\mathcal{T}_2$ can be potentially accelerated by first learning a different task $\mathcal{T}_1$. The potential speedup $\tau_{0-2} - (\tau_{0-1} + \tau_{1-2})$ will depend upon the similarity between the tasks, which in this work are minimally described by how their inputs co-vary, as well as how inputs vary with targets. **B.** Linear RNN architecture. Only $W_x$ and $W_y$ are trained, and $W_h$ are fixed, to facilitate closed-form solutions for their training dynamics. **C.** We study the problem of RNN parameter learning in a rotated reference frame that demixes and aligns the input and output singular "modes" with eigenmodes of the recurrent network. This allows for factorized learning where each mode of the input and output utilizes individual modes of the network.

on another task "1", then switches to the target. In particular, we want to understand under what circumstances and for what kind of pairs of tasks (each with a similar geometric alignment of their input-input and input-output covariance functions) that training the sequence offers speed benefits relative to training the target alone $\tau_{0-2} > \tau_{0-1} + \tau_{1-2}$ (Fig. 1A).

## 2.2 LEARNING DYNAMICS

We start by re-deriving a core result from Proca et al. (2025), which is the closed form expressions for the learning time course of the $W_x$ and $W_y$, for a fixed $W_h$. Learning the recurrent dynamics –either independently or jointly with the other parameters– does not afford simple closed form expressions, requiring more complex approximations. While focusing on input and output weights seems like a big simplification, the resulting parameter dynamics can nonetheless provide nontrivial insights into the multi-task learning process.

We sketch the key steps of the derivation in the main text, leaving the details to the Appendix. A key assumption in the optimization of $W_x$ and $W_y$ is that the primary axes of (co)variation of the inputs and outputs provided by $\Sigma_{x_t y}$ and $\Sigma_{x_t x'_t}$ can be aligned to the eigenmodes of the RNN, such that parameter learning can happen in a factorized manner (see Proca et al. (2025) for a discussion on the conditions when such alignment is possible). The geometry (*i.e.*, SVD axes) of $\Sigma_{x_t y}$ and $\Sigma_{x_t x'_t}$ are fixed over time, but their singular values can have time dependence. These "modes" of inputs and outputs (*i.e.*, formally the columns and rows of $W_x$ and $W_y$ in a rotated reference frame, respectively) require rotating the loss function based upon the Schur decomposition of $W_h = U_h H_h U_h^\top$, as well as the singular value decomposition (SVD) of the two task covariances

$$\boldsymbol{\Sigma}_{x_t x_{t'}} = \mathbb{E}[\boldsymbol{x}_t \boldsymbol{x}_{t'}^\top] \approx \sum_p^P \boldsymbol{x}_{p,t} \boldsymbol{x}_{p,t'}^\top = \boldsymbol{U}_{xx} \boldsymbol{S}_{x_t x_{t'}} \boldsymbol{U}_{xx}^\top \tag{5}$$

$$\boldsymbol{\Sigma}_{x_t y} = \mathbb{E}[\boldsymbol{x}_t \boldsymbol{y}_T^\top] \approx \sum_p^P \boldsymbol{x}_{p,t} \boldsymbol{y}_{p,T}^\top = \boldsymbol{U}_{xy} \boldsymbol{S}_{x_t y} \boldsymbol{V}_{xy}^\top. \tag{6}$$

The transformation of the problem in this rotated space is explained graphically in Fig. 1C. The input and output sequences are transformed in a way that recasts the problem into a factorized RNN consisting of a collection of parallel input-integrate-output channels indexed by $\alpha$, parametrized by a new set of parameters, $a_\alpha$, $b_\alpha$ and $c_\alpha$. These modes are the columns of $W_x$ ($a_\alpha$) eigenvectors of $W_h$ ($b_\alpha$), and rows of $W_y$ ($c_\alpha$) in this factorized reference frame, and we refer them as the input, recurrent, and output connectivity modes, respectively. For a small enough learning rate, one can

write learning dynamics for the input and output modes as:

$$\frac{\partial \boldsymbol{a}_\alpha}{\partial \tau} = \sum_{t,t'} b_\alpha^{(T-t)} \boldsymbol{c}_\alpha \left[ s_{yx_t}^\alpha - b_\alpha^{(T-t')}(\boldsymbol{c}_\alpha \cdot \boldsymbol{a}_\alpha)s_{x_t x_{t'}}^\alpha \right] \tag{7}$$

$$\frac{\partial \boldsymbol{c}_\alpha}{\partial t} = \sum_{t,t'} b_\alpha^{(T-t)} \boldsymbol{a}_\alpha \left[ s_{yx_t}^\alpha - b_\alpha^{(T-t')}(\boldsymbol{c}_\alpha \cdot \boldsymbol{a}_\alpha)s_{x_t x_{t'}}^\alpha \right]. \tag{8}$$

where $s_{x_t x_{t'}}^\alpha$ and $s_{yx_t}^\alpha$ reflect the time-varying singular values of the two covariance functions which define the task.

In general, this problem is not well-posed for any initial conditions of $\boldsymbol{a}_\alpha$ and $\boldsymbol{c}_\alpha$; however, under the special assumption that the $\boldsymbol{a}_\alpha$ and $\boldsymbol{c}_\alpha$ are initialized onto the same mode of an orthogonal basis with coefficients $a_\alpha$ and $c_\alpha$, a closed form solution for their product captures the learning dynamics:

$$a_\alpha(\tau)c_\alpha(\tau) = \frac{1}{\left[ \frac{1}{(a_\alpha(0)c_\alpha(0))} - \frac{\beta_{xx}^\alpha}{\beta_{yx}^\alpha} \right] e^{-2\tau \beta_{yx}^\alpha / \gamma} + \frac{\beta_{xx}^\alpha}{\beta_{yx}^\alpha}}, \tag{9}$$

where $\tau$ is the parameter update timestep, $\gamma$ is the inverse of the learning rate, and the effect of the recurrent network strength and singular values of the task covariances is captured by $\beta_{yx}^\alpha$ and $\beta_{xx}^\alpha$:

$$\beta_{yx}^\alpha = \sum_t^T b_\alpha^{(T-t)} s_{yx_t}^\alpha, \qquad\qquad \beta_{xx}^\alpha = \sum_{t,t'}^T b_\alpha^{(2T-t-t')} s_{x_t x_{t'}}^\alpha. \tag{10}$$

We refer to the $\beta$ terms as *recurrence-weighted singular values* (RWSV), as they account for the effect of $\boldsymbol{\Sigma}_{xx}$ and $\boldsymbol{\Sigma}_{yx}$, weighted by the effect of recurrent dynamics. Eq. 9 describes training for a single mode $\alpha$, so there will be equivalent expressions for each of the $\alpha = \{1, 2, .. \min[N_x, N_y]\}$ modes. This important result dictates the time course of parameter learning, and the optimal task solution. In the following section, we will utilize this expression to derive our core results that relate training time to task structure in CL.

## 2.3 THE IMPORTANCE OF TASK SIMILARITY FOR CURRICULUM LEARNING

Our primary goal is to understand the conditions in which learning an intermediate task accelerates learning of a target task. As a minimal example, we consider two tasks in sequence, which are defined by their covariance matrices: $\mathcal{T}_k = \{\boldsymbol{\Sigma}_{xx}^{(k)}, \boldsymbol{\Sigma}_{xy}^{(k)}\}, k = 1, 2$. Moving forward we denote the product of input and outputs mode coefficients as $ac = u$. Starting from initial conditions $u_0$, consider the optimization time needed until $u$ is within a small $\epsilon$ tolerance of the optimal solution for task $\mathcal{T}_1$, denoted by $u^{*(1)}$. To begin, the optimal solution is found for $t \to \infty$ in Eq. 9,

$$u^{*(k)} = \beta_{yx}^{(k)} / \beta_{xx}^{(k)} = \frac{\sum_i^T b^{T-t} s_{yx_t}^{(k)}}{\sum_{t,t'}^T b^{2T-t-t'} s_{x_t x_{t'}}^{(k)}}. \tag{11}$$

We note that in the special case of constant singular values and perfectly stable dynamics ($b = 1$), this recapitulates the results in Saxe et al. (2014) (Appendix A.4.4).

Rearranging Eq. 9, we can solve for the amount of training required to reach a convergence criterion $u(\tau) = (1 - \epsilon)u^{*(1)}$, where the precision of the final solution relative to the optimum $u^{*(1)}$ is determined by parameter $\epsilon$:

$$t_{i \to 1} = \frac{\gamma}{2\beta_{yx}^{(1)}} \left( \log \left| \frac{u^{*(1)}}{u_0} - 1 \right| - \log \left| \frac{\epsilon}{1 - \epsilon} \right| \right). \tag{12}$$

Thus, the training time can be separated into the relationship between optimal solutions and initial conditions, as well as the desired error tolerance. Given this, it is straightforward to calculate the time to optimize along a sequence of two tasks $\mathcal{T}_1$ and $\mathcal{T}_2$:

$$t_{i \to 2} = t_{i \to 1} + t_{1 \to 2} \tag{13}$$

$$= \frac{\gamma}{2\beta_{yx}^{(1)}} \left( \log \left| \frac{u^{*(1)}}{u_0} - 1 \right| - \log \left| \frac{\epsilon^{(1)}}{1 - \epsilon^{(1)}} \right| \right)$$

$$+ \frac{\gamma}{2\beta_{yx}^{(2)}} \left( \log \left| \frac{u^{*(2)}}{(1 - \epsilon^{(1)})u^{*(1)}} - 1 \right| - \log \left| \frac{\epsilon^{(2)}}{1 - \epsilon^{(2)}} \right| \right), \tag{14}$$

where we have denoted the error tolerance for each task as $\epsilon^{(k)}$. Importantly, this training time only holds if the geometry of $\mathcal{T}_1$ is equivalent to $\mathcal{T}_2$, meaning that the the SVD eigenvectors for the task covariances are the same in both tasks. Otherwise, after training on $\mathcal{T}_1$, the initial conditions would not lie in an orthogonal basis set by the eigenvectors of $\mathcal{T}_2$, and there would be cross-mode contributions during training (numerical results of this scenario provided in Appendix A.4.2 ).

Our primary result determines the conditions under which training on $\mathcal{T}_1$ offers a speedup when learning a task $\mathcal{T}_2$ with equivalent task geometry,

$$t_{i \to 2} > t_{i \to 1} + t_{1 \to 2}. \tag{15}$$

Expanding Eq. 15 highlights the relationships between task singular values, task accuracy, and training speed:

$$\log \left| \frac{u^{*(2)}}{u_0} - 1 \right| + \frac{\beta_{yx}^{(2)}}{\beta_{yx}^{(1)}} \log \left| \left( \frac{\epsilon^{(1)}}{1 - \epsilon^{(1)}} \right) \left( \frac{u_0}{u^{*(1)} - u_0} \right) \right| - \log \left| \frac{1}{(1 - \epsilon^{(1)})} \frac{\beta_{yx}^{(2)}}{\beta_{yx}^{(1)}} \frac{\beta_{xx}^{(1)}}{\beta_{xx}^{(2)}} - 1 \right| > 0 \tag{16}$$

Eq. 16 details the conditions under which there will be a speedup in first training on $\mathcal{T}_1$. The different aspects of task structure that drive faster learning are nonlinearly related, so to gain insight we examine each term individually to hypothesize what it implies about relative task structure in CL. The first term simply implies that –provided the initial conditions are suitably small– there will be a speedup, which does not relate task structure to training time. The second term does relate input-target singular values across tasks, and suggests that when $\beta_{yx}^{(1)} > \beta_{yx}^{(2)}$, CL sees faster training. The third term also shows this (provided that $\beta_{xx}^{(1)} = \beta_{xx}^{(2)}$), as well as the inverse relationship that CL is faster when $\beta_{xx}^{(1)} < \beta_{xx}^{(2)}$ (also provided $\beta_{yx}^{(1)} = \beta_{yx}^{(2)}$). Finally, we re-write the 3rd term with respect to the optimal solutions to show a surprising result that training intermediate tasks to potentially low accuracies can be beneficial

$$-\log \left| \frac{u^{*(2)}}{(1 - \epsilon) u^{*(1)}} - 1 \right| > 0. \tag{17}$$

There is a singularity in this expression whenever $\mathcal{T}_1$ has been optimized to exactly be the solution to task $\mathcal{T}_2$, which can produce a CL speedup when it is in the neighborhood of this singularity. Interestingly, depending on the magnitude of the two optimal solutions, this speedup can occur for small accuracy on the first task.

These regimes from eq. 17 are general conditions where CL is worthwhile, and for a given task type they have intuitive and practical explanations. In short, these conditions spell out what makes an intermediate task "easier" than the second one. For example, in our integration tasks studied here, our theory predicts that when inputs strongly correlate with the target output, it is easier than a weakly correlated task and will help training. This is a consequence of our first observation that $\beta_{yx}^{(1)} > \beta_{yx}^{(2)}$. Additionally, if inputs are highly similar to one another then the integration problem reduces instead to simply scaling a single input to a target value, a much "easier" problem than full integration of a time-varying signal. This is a consequence of $\beta_{xx}^{(1)} < \beta_{xx}^{(2)}$, which occurs for weaker overall input covariance strength, as well as for very temporally correlated inputs.

In summary our theory predicts three broad effects on CL speed that are related to task structure that have practical benefits, and relate to intuitive ideas of task "easiness" commonly found in CL sequences: 1) $\beta_{yx}^{(1)} > \beta_{yx}^{(2)}$, 2) $\beta_{xx}^{(1)} < \beta_{xx}^{(2)}$, and 3) training to suboptimal accuracies on an intermediate task can be beneficial. We next turn to numerical simulations to validate our theory, as well as to explore how the strength and temporal correlations of task covariances support these effects.

## 3 NUMERICAL VALIDATION

In the above section we showed analytically that recurrence weighted singular values can drive CL, which we verify here. We examine the numerical optimization of two tasks in sequence, and compare that training time to a second task. For ease of visualization and to demonstrate core features of the theory, we study networks with a single input and output channel. Additionally, because Eq. 16

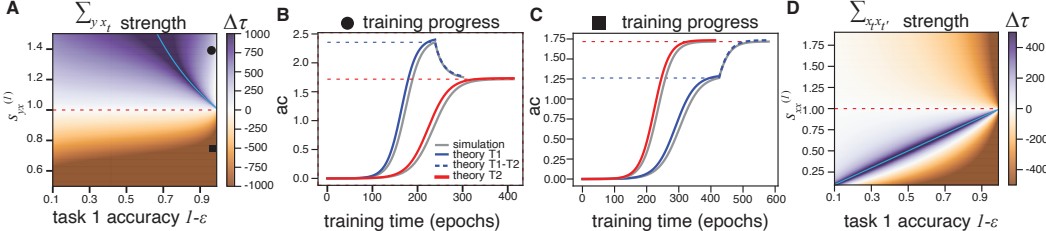

Figure 2: **A.** Phase portrait of difference in training time between direct $\mathcal{T}_2$ training and a curriculum of $\mathcal{T}_1$ (to accuracy $1 - \epsilon$) followed by task $\mathcal{T}_2$. $\mathcal{T}_1$ accuracy and singular value strength or $\Sigma_{yx_t}$ were modulated. Dotted red line shows singular value of $\mathcal{T}_2$. Cyan denotes singularity when $\mathcal{T}_1$ solution corresponds to $\mathcal{T}_2$ optimum, requiring no training time. **B-C.** Predicted vs. numerical optimization training trajectories for individual parameter settings, denoted by square and circle in panel **A**. **D** Similar phase portraits as in **A**, but for modulating singular value of $\Sigma_{x_t, x_t'}$.

nonlinearly relates multiple parameters, we will study different aspects of task structure individually, while also varying task accuracy. In the simulations below we assume the accuracy of task $\mathcal{T}_2$ is $1 - \epsilon^{(2)} = 0.99$, and we set the recurrence mode $b = 0.96$ to ensure ideal RNN performance when comparing to numerical optimization, while still capturing the effects of recurrence. In all cases, networks contained 128 hidden units, trials were 50 timesteps long, and numerical comparisons to theory trained with batches of 1000 samples.

### 3.1 TASK COVARIANCE STRENGTH

We first focus on scenarios in which there is no temporal correlation in the task covariances, and only the strength of covariance can modulate training speed. When examining the input-to-target covariance, our theory predicts that intermediate tasks with larger $\Sigma_{yx_t}$ singular values will be beneficial, so to isolate this effect we studied a set of tasks no temporal correlation ($\Sigma_{x_t, x_t'} = a\delta_{t,t'}$). We used Eq. 12 to compute the training time for $\mathcal{T}_2$, as well as Eq. 13 for the training time for learning $\mathcal{T}_1$ to accuracy $1 - \epsilon$, followed by learning $\mathcal{T}_2$. We then examined the difference in training time for a range of $\mathcal{T}_1$ accuracy and singular value amplitudes for $\mathcal{T}_1$ as a phase portrait in Fig.2A. Sample numerical training trajectories compared to theory are provided in Fig. 2B-C.

We found that our hypothesis from Eq. 16 holds, where first training on tasks with larger singular values led to faster training. Practically, this implies that tasks with inputs that are more saliently related to the targets are ideal candidates for curricula. Somewhat surprisingly, this means that tasks that tune input and output weights to initially larger values aid in learning later tasks with smaller weights. We additionally see in Fig. 2A that training $\mathcal{T}_1$ to even modest accuracies still improve performance. where a larger range of accuracies is beneficial when $\mathcal{T}_1$ has relatively larger singular values. This is due to the singularity in training time when the final solution for $\mathcal{T}_1$ is near the the optimal solution for $\mathcal{T}_2$, which creates a basin of parameter values that provide a CL speedup (2A, cyan line). We next examined the variance of inputs in the same manner in Fig. 2D. Here, we found that the relationship in training speed was generally flipped as expected, with $\mathcal{T}_1$ tasks containing weaker input variances being more beneficial to training speed. Sample learning trajectories and numerical comparison to theory are provided in the appendix (Supp. Fig. 5).

### 3.2 TEMPORAL CORRELATIONS IN TASK COVARIANCES

We next turn to investigating how changes to the temporal structure of the task covariances can facilitate CL. To see how the temporal properties of the task can support this, we study correlated inputs generated by an AR1 process as

$$x_t = Kx_{t-1} + w_t, \tag{18}$$

where $w_t \sim \mathcal{N}(0, \Sigma_0)$ is white noise with covariance $\Sigma_0$, and $K$ defines the strength of the temporal correlation. For this process, input covariances depend only upon the lag between time-points (Fig.

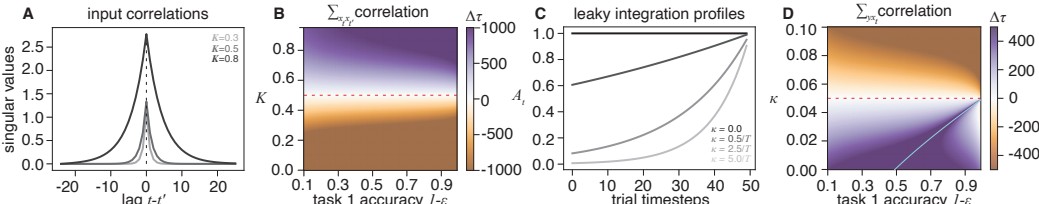

Figure 3: Temporal correlations affect CL. **A.** We studied inputs drawn from AR1 process in which strength of singular values is lag-dependent, given by strength $K$ (see main text). **B.** Phase portrait showing difference in training time for CL sequence vs. direct target task training. Accuracy and correlation of $\mathcal{T}_1$ were varied relative to $\mathcal{T}_2$ being trained to 99% accuracy ($K$ for $\mathcal{T}_2$, dotted red line). **C.** Integration profile for leaky integration tasks. Time-dependent $\Sigma_{yx_t}$ corresponds to targets that perform leaky integration with exponentially decaying profiles with timescale $\kappa$. **D.** Phase portrait as in **B**, but for varying time $\kappa$. Cyan denotes singularity when $\mathcal{T}_1$ solution corresponds to $\mathcal{T}_2$ optimum, requiring no training time.

3A) [3]. Here we consider integration tasks that do not simply perform perfect integration across all time, but are instead leaky integration tasks that weight later time points in a trial

$$y_t = \sum_t^T A_t x_t, \quad A_t = A_0 e^{-\kappa(t-T)} \tag{19}$$

where $A_0$ is the $N_y \times N_x$ matrix that mixes inputs to output channels, and $\kappa$ is the decay of the integration profile (Fig. 3C).

We again calculated the difference in time to train a target task $\mathcal{T}_2$ vs. training intermediate task $\mathcal{T}_1$ first, followed by $\mathcal{T}_2$, but with varying accuracy and temporal properties $K$ and $\kappa$ of the task covariances. When modulating input correlations $K$ (Fig. 3A), we find that stronger correlations in the inputs of $\mathcal{T}_1$ improve training speed (Fig. 3B). We next looked at the tradeoffs in task accuracy of task 1, and the timescale $\kappa$ of its temporal integration profile for leaky integration tasks (Fig. 3C). We find that intermediate tasks with longer integration windows lead to faster training on task $\mathcal{T}_2$. This is a scenario where determining what constitutes an "easy" task is less clear, but that is easily explained by our CL theory. Integrating over longer timescales would conventionally be thought of as more difficult (*e.g.*, requiring longer time horizons), but larger integration profiles produce a larger $\beta_{yx}^{(1)}$ (Eq. 10), which our theory predicts will produce an increase in training speed for CL. We again see evidence of suboptimal task 1 accuracy providing a speedup because it places initial conditions for task 2 training near a singularity (Fig. 3D, cyan line). Sample learning trajectories and numerical comparison to theory are provided in the appendix (Supp. Fig. 5).

In summary, we find that the time-dependent aspects of task covariances are an equally important dimension that can predict the success of CL. In the final section, we investigate if the insights from our theory of CL in linear RNNs will generalize once we relax assumptions about the network architecture.

### 3.3 NONLINEAR RNNs

Finally, we wished to see if the insights found in our linear RNN analysis would hold in a more practical scenario. So we performed the same CL studies, using the same integration tasks, but with RNNs containing a ReLU nonlinearity. For individual RNNs, we compared the training time for networks that had either a linear or ReLU activation function, and we investigated four different parameter regimes that characterize the main aspects of task covariances (Fig. 4). Without the ability to generate theoretical predictions for nonlinear networks, we instead focused on whether or not the same qualitative principles identified in our linear theory would hold in nonlinear networks. While there are numerical differences in the optimization time, we found evidence that the same qualitative

---

[3]correlations that are purely lag-dependent hold only in the infinite-time limit, and we account for finite time correlations when we calculate $\Sigma_{x_t x_t'}$.

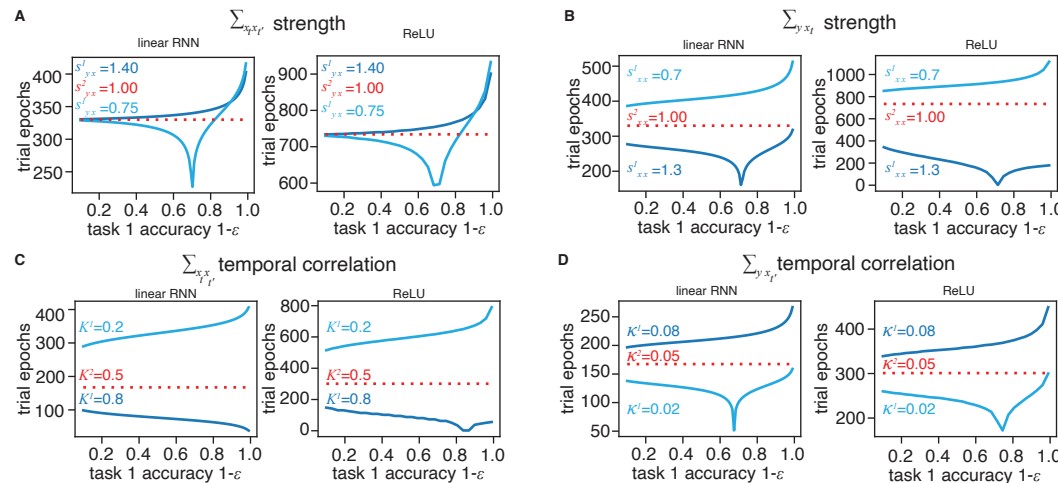

Figure 4: CL effects in nonlinear RNNs compared linear RNN theoretical predictions. Total training times for individual RNNs across a range of $\mathcal{T}_1$ accuracies for two different task covariance parameter settings: Red lines denote directly training on $\mathcal{T}_2$ to 99% accuracy and blue lines denote a CL sequence training on $\mathcal{T}_1$ first, with either a larger (dark blue) or smaller (light blue) parameter. Parameter values for each scenario are provided as legends. **A-B** Modulating input-target covariance strength as in section 3.1. **C-D** Modulating temporal correlations in task covariance as in section 3.2

trends seen for linear RNNs can hold even for nonlinear networks, meaning the the relative amplitude of task covariance strength and temporal correlations between two tasks appears to hold. Sample learning trajectories are provided in the appendix (Supp. Fig. 5). Finally, we also studied additional extensions beyond our theory where its qualitative trends still hold, including tasks with mismatched task geometry (Supp. Fig. 6), as well as jointly training recurrent and input/output weights (Supp. Fig. 8).

# 4 DISCUSSION

Our work set out to provide a theoretical understanding of the benefits of curriculum learning for speeding up learning in a target task. To make progress, we distilled this goal into a concrete mathematical question: what aspects of similarity between two tasks support faster learning in linear RNNs? Building upon recent theoretical results Proca et al. (2025), we derived how the strength and temporal structure of the covariances between inputs and between inputs and outputs shape pretraining efficiency. Our theory predicted three primary drivers of CL success: 1) stronger singular values in input-target covariances and larger target integration windows in the first task, 2) weaker singular values in the input-input covariance and more temporally correlated inputs; In our example system we showed how these relationships comported with conventional ideas about task 'easiness.' Finally we found that 3) training speed can benefit from suboptimal task accuracy in the first task. This was not simply due to avoiding a sunk cost in over-training on the first task, but rather an effect of strong overlap between task 1 solutions at low accuracy and the target task solution.

While our general approach follows recent results on the learning dynamics of input and output weights in linear RNNs (Proca et al., 2025), it expands technically on them in several important ways. First, unlike previous work we had to take into account the temporal dependencies in input and outputs. This is something that can be avoided with appropriate re-parametrization when considering single tasks, but needs to be considered explicitly once multiple tasks are analyzed together in the same coordinate system. The second technical contribution is directly deriving time to convergence for single tasks and sequences of tasks. This advance enabled us to build explicit phase plane analyses for what kind of tasks lead to learning speedups across a range of scenarios, the results of which we were able to confirm numerically.

Our results spell out the key properties about task relationships that allow for faster training, and an ultimate goal of this work is to provide simple heuristics for how to harness these relationships to build straightforward stopping criteria on pre-training tasks. Our theoretical work demonstrated the existence of a singularity condition for training time improvement (eq. 17), and through numerical simulations we found that this singularity provided a broad range of support for nearby solutions to have tangible training speedups (Fig. 2A,2D, 3D). The nonlinear relationship between relative task covariances and training speedup suggests a general guiding strategy for when to stop training on an initial task: the larger that differences in task covariances, the earlier one should stop training on the first task. In particular, the mean-squared error tasks studied here have sigmoidal training trajectories (eq. 9) and unique optima (eq. 11), which provides a simple diagnostic for when to stop training on task 1: Monitoring the second task for a steep decrease in its loss, following by a saturation hints that training on task 1 has placed you in the neighborhood of the optimal solution for task 2. This is highly specific to the nature of the loss function for this task, and different tasks will have different signatures. Future work aims to determine such practical stopping heuristics in other task classes.

The main limitations of our current approach is the restriction to similar pairs of tasks with common structure, and the focus on the learning of input and output parameters. First, to be able to make mathematical progress, we had to assume that sequences of tasks maintain the same general "task geometry." The next natural step would be to relax this constraint by investigating the time required to rotate a linear system into a factorized training regime (Fig. 1C), perhaps by taking advantage of recent work demonstrating a natural alignment effect into such diagonalized regimes (Atanasov et al., 2021). As a counterpart for this focus on alignment, one could perhaps embrace the inherent mixing of network modes to study tasks with compositional structure, which combine computations from separate modes to perform new ones. This is an interesting arena to study CL, as there has been evidence that CL is required for complex compositional tasks in RNNs (Hocker et al., 2025; Krueger & Dayan, 2009), and would complement existing efforts to characterize compositional pretraining in feed-forward networks (Lee et al., 2024).

With respect to the second main limitation, here we restricted our analysis to training input and output weights in RNNs with predefined recurrence. While this is certainly restrictive, it is nonetheless directly applicable to transfer-learning scenarios when the network's internal representations are reused, while input/output weights are adapted to novel inputs and targets (Pan & Yang, 2009). There is also a rich body of numerical results in computational neuroscience that examine how banks of dynamical motifs can be reused and composed to perform complex tasks Driscoll et al. (2024). Going forward, it would be important to jointly study the effect of recurrence in shaping task similarity and influencing the outcomes of curriculum learning. Incorporating recurrence into our analysis is potentially possible, as there is already a theoretical basis for learning recurrence in the domain of computations at long timescales (Schuessler et al., 2020).

Finally, here we have mainly focused on learning speed as a metric of success for CL, at the detriment of other benefits such as robustness of the solution, generalization quality, or sensitivity to noise. These other factors have important practical relevance and will need to be considered in subsequent analyses. The ultimate goal with a theoretical description like ours is to inform practical machine learning problems. Given the recent demonstrations that even highly simplified mathematical analyses can still carry insight into mathematically intractable but practically relevant scenarios (Liu et al., 2024), we hope that our approach can make an impact throughout the the breadth of the CL ecosystem (Soviany et al., 2022).

### REPRODUCIBILITY STATEMENT

In order to facilitate reproducibility of our results we have included a full derivation of the theory in the Appendix. We have also provided details for the simulation and training of RNNs that lead to the results. All code for generating the results in the manuscript will be provided at the time of publication.

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

## A  APPENDIX

### A.1  EXTENDED DERIVATION OF LEARNING DYNAMICS

In this section we derive Eq. 9. Our approach is based on Proca et al. (2025) and Saxe et al. (2014), and considers a slightly broader range of tasks with fewer constraints on the task geometry. While we ultimately consider a regime that is similar to Proca et al. (2025), we aim to keep the derivation as general as possible and highlight when assumptions are required to yield tractable analytical solutions. We hope that this exposes future directions for the theory of learning in RNNS.

We begin with a linear RNN of the form

$$\boldsymbol{h}_t = \boldsymbol{W}_h \boldsymbol{h}_{t-1} + \boldsymbol{W}_x \boldsymbol{x}_t \tag{20}$$

$$\boldsymbol{y}_t = \boldsymbol{W}_y \boldsymbol{h}_t, \tag{21}$$

which maps time-varying inputs $\boldsymbol{x}_t \in \mathbb{R}^{N_x \times 1}$ into a network state $\boldsymbol{h}_t \in \mathbb{R}^{N_h \times 1}$, read out into outputs $\boldsymbol{y}_t \in \mathbb{R}^{N_y \times 1}$. The learnable parameters of the network include the recurrent weight matrix $\boldsymbol{W}_h \in \mathbb{R}^{N_h \times N_h}$, input matrix $\boldsymbol{W}_x \in \mathbb{R}^{N_h \times N_x}$, and output matrix $\boldsymbol{W}_y \in \mathbb{R}^{N_y \times N_h}$. The RNN will be optimized to perform on task pulled from a family of leaky integration tasks, where inputs $\boldsymbol{x}_{1:T}$ are integrated over time with different linear filters to yield target outputs, $\hat{\boldsymbol{y}}_T$, at the end of the trial, $T$. The loss over a batch of $P$ trials for this single output scenario of generating a target $\boldsymbol{y}$ is given as

$$\mathcal{L} = \frac{1}{2} \sum_p^P \|\boldsymbol{y}_p - \hat{\boldsymbol{y}}_{T,p}\|^2. \tag{22}$$

Starting from initial state $\boldsymbol{h}_0 = \boldsymbol{0}$, the network dynamics evolve as

$$\boldsymbol{h}_t = \sum_{i=1}^t \boldsymbol{W}_h^{t-i} \boldsymbol{W}_x \boldsymbol{x}_i, \tag{23}$$

which allows the loss to be rewritten as

$$\mathcal{L} = \sum_{t,t'=1}^T \frac{1}{2} \operatorname{Tr} \left[ \boldsymbol{W}_y \boldsymbol{W}_h^{T-t} \boldsymbol{W}_x \boldsymbol{\Sigma}_{x_t x_{t'}} \boldsymbol{W}_x^\top \boldsymbol{W}_h^{T-t'\top} \boldsymbol{W}_y^\top - \boldsymbol{W}_y \boldsymbol{W}_h^{T-t} \boldsymbol{W}_x \boldsymbol{\Sigma}_{x_t y} \right] + \text{const.} \tag{24}$$

The autocorrelation function of the inputs, $\boldsymbol{\Sigma}_{x_t x_t'} = \mathbb{E}[\boldsymbol{x}_t \boldsymbol{x}_{t'}^\top]$, together with the cross-correlation between time-varying input and targets, $\boldsymbol{\Sigma}_{x_t y} = \mathbb{E}[\boldsymbol{x}_t \boldsymbol{y}_T^\top]$ fully specify an instance of the task. Different tasks will have different $\boldsymbol{\Sigma}_{x_t x_t'}$ and $\boldsymbol{\Sigma}_{x_t y}$, with varying degrees of overlap.

The loss in Eq. 24 depends on the learnable parameters, as well as data-averaged task covariances $\Sigma_{x_t x_{t'}}$ that describe how inputs co-vary over time and input dimensions, as well as input-target co-variances $\Sigma_{x_t y}$ that describe how inputs co-vary with target values. These covariances have singular value decompositions (SVD) given by

$$\boldsymbol{\Sigma}_{x_t x_{t'}} = \mathbb{E}[\boldsymbol{x}_t \boldsymbol{x}_{t'}^\top] \approx \sum_p^P \boldsymbol{x}_{p,t} \boldsymbol{x}_{p,t'}^\top = \boldsymbol{U}_{xx} \boldsymbol{S}_{x_t x_{t'}} \boldsymbol{U}_{xx}^\top \tag{25}$$

$$\boldsymbol{\Sigma}_{x_t y} = \mathbb{E}[\boldsymbol{x}_t \hat{\boldsymbol{y}}_T^\top] \approx \sum_p^P \boldsymbol{x}_{p,t} \hat{\boldsymbol{y}}_{p,T}^\top = \boldsymbol{U}_{xy} \boldsymbol{S}_{x_t y} \boldsymbol{V}_{xy}^\top \tag{26}$$

Consistent with the previous work (Proca et al., 2025), we make additional assumption regarding the form of the task covariances: 1) we assume a static "task geometry," meaning that that the SVD axes $(\boldsymbol{U}_{xx}, \boldsymbol{U}_{xy}, \boldsymbol{V}_{xy})$ are constant over time, which implies a constant input-to-output mapping during the task. Unlike previous work, we do not assume fully whitened inputs here. When considering learning a sequence of tasks this assumption would be too restrictive: while it is possible to fully whiten a target task, the corresponding coordinate system will not necessarily whiten inputs for the pretraining tasks.

Next, we recast the loss function in a rotated space that couples singular values of $\boldsymbol{\Sigma}_{x_t y}$ with recurrent modes provided by a Schur decomposition as $\boldsymbol{W}_h = \boldsymbol{U}_h \boldsymbol{H}_h \boldsymbol{U}_h^\top$, where $\boldsymbol{H}_h$ is upper-triangular for non-normal dynamics, and diagonal for normal dynamics. Rotating the input and output weights as $\boldsymbol{W}_x = \boldsymbol{U}_h \tilde{\boldsymbol{W}}_x \boldsymbol{U}_{xx}^\top$, $\boldsymbol{W}_y = \boldsymbol{V}_{xy} \tilde{\boldsymbol{W}}_y \boldsymbol{U}_h^\top$, the loss function becomes

$$\mathcal{L} = \sum_{t,t'}^{T} \frac{1}{2} \mathrm{Tr}\left[ \tilde{\boldsymbol{W}}_y \boldsymbol{H}_h^{T-t} \tilde{\boldsymbol{W}}_x \boldsymbol{S}_{x_t x_{t'}} \tilde{\boldsymbol{W}}_x^\top \boldsymbol{H}_h^{T-t'\top} \tilde{\boldsymbol{W}}_y^\top \right] - \mathrm{Tr}\left[ \tilde{\boldsymbol{W}}_y \boldsymbol{H}_h^{T-t} \tilde{\boldsymbol{W}}_x (\boldsymbol{U}_{xx}^\top \boldsymbol{U}_{xy}) \boldsymbol{S}_{x_t y} \right]. \tag{27}$$

We make a further assumption here that that $\boldsymbol{U}_{xx}^\top \boldsymbol{U}_{xy} = \boldsymbol{I}$, which holds only for $\boldsymbol{U}_{xx} = \boldsymbol{U}_{xy}$. This implies a connection between the axes of the input covariability and the SVD modes of the input-to-output mapping, which is that the directions of variability in the inputs must be aligned with the primary SVD modes of $\boldsymbol{\Sigma}_{x_t y}$. [4] At this stage, we do not assume that input and output matrices are naturally aligned to the network and singular values modes, meaning $\tilde{\boldsymbol{W}}_x$ and $\tilde{\boldsymbol{W}}_y$ are not assumed to be diagonal. We will implement this in practice when by choosing a privileged set of initial conditions for training, but our derivation does not require this.

We restrict the learning dynamics to how input and output parameters to the network update over learning, as the learning dynamics for recurrent weights do not have analytical solutions without introducing approximations. By denoting the learning trajectory by a variable $\tau$, these updates are given by

$$\frac{\partial \tilde{\boldsymbol{W}}_x}{\partial \tau} = -\frac{\partial \mathcal{L}}{\partial \tilde{\boldsymbol{W}}_x} = \sum_{t,t'} \boldsymbol{H}_h^{T-t\top} \tilde{\boldsymbol{W}}_y^\top \boldsymbol{S}_{yx_i} - \boldsymbol{H}_h^{T-t'\top} \tilde{\boldsymbol{W}}_y^\top \tilde{\boldsymbol{W}}_y \boldsymbol{H}_h^{T-t} \tilde{\boldsymbol{W}}_x \boldsymbol{S}_{x_t x_{t'}}, \tag{28}$$

$$\frac{\partial \tilde{\boldsymbol{W}}_y}{\partial \tau} = -\frac{\partial \mathcal{L}}{\partial \tilde{\boldsymbol{W}}_y} = \sum_{t,t'} \boldsymbol{S}_{yx_t} \tilde{\boldsymbol{W}}_x^\top \boldsymbol{H}_h^{T-t\top} - \tilde{\boldsymbol{W}}_y \boldsymbol{H}_h^{T-t} \tilde{\boldsymbol{W}}_x \boldsymbol{S}_{x_t x_{t'}} \tilde{\boldsymbol{W}}_x^\top \boldsymbol{H}_h^{T-t'\top}. \tag{29}$$

We note that Eqs. 28-29 hold for both non-normal dynamics and normal dynamics. Moving forward, though, we restrict our attention to the case of normal dynamics (diagonal $H_h$).We also now make the same diagonalized matrix assumptions in Proca et al. (2025), which is that $\tilde{W}_x$ and $\tilde{W}_y$ have only diagonal entries. This yields update equations where $\boldsymbol{H}$ can combine

$$\frac{\partial \tilde{\boldsymbol{W}}_x}{\partial \tau} = -\frac{\partial \mathcal{L}}{\partial \tilde{\boldsymbol{W}}_x} = \sum_{t,t'} \boldsymbol{H}_h^{T-t} \tilde{\boldsymbol{W}}_y^\top \boldsymbol{S}_{yx_i} - \boldsymbol{H}_h^{2T-t'-t} \tilde{\boldsymbol{W}}_y^\top \tilde{\boldsymbol{W}}_y \tilde{\boldsymbol{W}}_x \boldsymbol{S}_{x_t x_{t'}} \tag{30}$$

$$\frac{\partial \tilde{\boldsymbol{W}}_y}{\partial \tau} = -\frac{\partial \mathcal{L}}{\partial \tilde{\boldsymbol{W}}_y} = \sum_{t,t'} \boldsymbol{S}_{yx_t} \tilde{\boldsymbol{W}}_x^\top \boldsymbol{H}_h^{T-t} - \tilde{\boldsymbol{W}}_y \boldsymbol{H}_h^{2T-t-t'} \tilde{\boldsymbol{W}}_x \boldsymbol{S}_{x_t x_{t'}} \tilde{\boldsymbol{W}}_x^\top. \tag{31}$$

Rather than track how updates for the entire weight matrices unfold under time, it is useful to consider how their columns and vectors, or "modes", of these matrices update over time Saxe et al. (2014); Proca et al. (2025). Specifically, we define the columns of $\tilde{\boldsymbol{W}}_x$ as $\boldsymbol{a}_\alpha$, and the rows of $\tilde{\boldsymbol{W}}_y$ as $\boldsymbol{c}_\beta$. The diagonal entries of $\boldsymbol{H}_h$ are given by $b_\alpha$ (the eigenvalues of $W_h$), and similarly $s_{x_t y}^\alpha$ and $s_{x_t x_{t'}}^\alpha$ are the diagonal entries of the task covariance matrices. The modes are then given as

$$\boldsymbol{a}(\tau) = \tilde{\boldsymbol{W}}_{x:,\alpha} = \sum_\alpha a_\alpha(\tau) \boldsymbol{r}_\alpha, \qquad \boldsymbol{b}(\tau) = \tilde{\boldsymbol{W}}_{y_{\alpha,:}} = \sum_\alpha b_\alpha(\tau) \boldsymbol{r}_\alpha, \tag{32}$$

---

[4] This was also noted in Saxe et al. (2014) Appendix

where $\{\boldsymbol{r}_\alpha\} \in R^{N_h \times 1}$ is a basis set of vectors for the modes.

By tracking the $\alpha$-th columns of $\tilde{W}_x$ and $\alpha$-th rows of $\tilde{W}_y$ in eqs. 30-31, we can express the update equations for the input and output modes:

$$
\begin{aligned}
\frac{\partial \boldsymbol{a}_\alpha}{\partial t} &= \sum_{t,t'} \sum_\gamma b_\gamma^{(T-t)} \boldsymbol{c}_\gamma s_{yx_i}^\gamma - b_\alpha^{(T-t)} b_\alpha^{(T-t')} (\boldsymbol{c}_\alpha \cdot \boldsymbol{a}_\alpha) s_{x_t x_{t'}}^\alpha \\
&= \sum_{t,t'} b_\alpha^{(T-t)} \boldsymbol{c}_\alpha \left[ s_{yx_t}^\alpha - b_\alpha^{(T-t')} (\boldsymbol{c}_\alpha \cdot \boldsymbol{a}_\alpha) s_{x_t x_{t'}}^\alpha \right] - \sum_{\gamma \neq \alpha} b_\gamma^{(T-t)} \boldsymbol{c}_\gamma s_{yx_t}^\gamma
\end{aligned}
\tag{33}
$$

$$
\frac{\partial \boldsymbol{c}_\alpha}{\partial t} = \sum_{t,t'} b_\alpha^{(T-t)} \boldsymbol{a}_\alpha \left[ s_{yx_t}^\alpha - b_\alpha^{(T-t')} (\boldsymbol{c}_\alpha \cdot \boldsymbol{a}_\alpha) s_{x_t x_{t'}}^\alpha \right] - \sum_{\gamma \neq \alpha} b^{\gamma^{(T-i)}} \boldsymbol{a}^\gamma s_{yx_i}^\gamma
\tag{34}
$$

Eqs. 33-34 contain contributions from their own mode $\alpha$, as well as cross term from other modes $\gamma$. Analytical solutions for this form are not generally tractable because of the contribution from all modes to learning, and so to address this we restrict our analysis to a special set of initial conditions to remove the cross-mode contribution. This is performed by initializing modes in a distinct set of non overlapping basis set vectors $\{\boldsymbol{r}_\alpha\} \in \mathbb{R}^{N_h \times 1}, \boldsymbol{r}_\alpha \cdot \boldsymbol{r}_\beta = \delta_{\alpha\beta}$: As has been shown previously Saxe et al. (2014), if $\boldsymbol{a}$ and $\boldsymbol{c}$ are initialized onto the same set of orthogonal modes $\{\boldsymbol{r}_\alpha\}$, then we can track the evolution of the coefficients on these modes, and importantly, any interaction terms among these modes are strictly zero with these initial conditions.

The update equations for the weighting coefficients are then given as

$$
\frac{\partial a_\alpha}{\partial \tau} = \sum_{t,t'} b_\alpha^{(T-t)} c_\alpha \left[ s_{yx_t}^\alpha - b_\alpha^{(T-t')} c_\alpha a_\alpha s_{x_t x_{t'}}^\alpha \right]
\tag{35}
$$

$$
\frac{\partial c_\alpha}{\partial t} = \sum_{t,t'} b_\alpha^{(T-t)} a_\alpha \left[ s_{yx_t}^\alpha - b_\alpha^{(T-t')} c_\alpha a_\alpha s_{x_t x_{t'}}^\alpha \right]
\tag{36}
$$

Because we ignored the mode cross terms, the updates in Eqs. 35-36 minimize an effective loss function, which can be seen by integrating them with respect to the mode coefficients:

$$
E = \sum_\alpha \sum_{t,t'}^T \left[ a_\alpha c_\alpha b_\alpha^{T-t} s_{x_t x_{t'}}^{\alpha^{1/2}} - s_{yx_t}^\alpha s_{x_t x_t'}^{\alpha^{-1/2}} \right] \left[ a_\alpha c_\alpha b_\alpha^{T-t'} s_{x_t x_{t'}}^{\alpha^{1/2}} - s_{yx_t}^\alpha s_{x_t x_{t'}}^{\alpha^{-1/2}} \right].
\tag{37}
$$

The product $a_\alpha c_\alpha$ has symmetry condition of this energy ($a_\alpha c_\alpha = [a_\alpha/k][c_\alpha k]$), which guarantees an invariance condition $a^2 = c^2$ Saxe et al. (2014). Moving forward, we omit the $\alpha$ index unless it is strictly necessary. Introducing a collective network parameter $u = ac$, the collective update equation follows a similar functional form

$$
\frac{\partial u}{\partial t} = \frac{\partial a}{\partial \tau} c + a \frac{\partial c}{\partial \tau} = 2 \sum_{i,j} b^{(T-t)} u \left[ s_{yx_t} - b^{(T-t')} u s_{x_t x_{t'}} \right]
\tag{38}
$$

where we used the equivalence $a^2 = c^2$. Eq. 38 is a separable differential equation with a closed form solution. To simplify notation, we collect the effect of recurrence and task covariances into terms $\beta_{yx}$ and $\beta_{xx}$

$$
\beta_{yx}^\alpha = \sum_t^T b_\alpha^{(T-t)} s_{yx_t}^\alpha, \qquad\qquad \beta_{xx}^\alpha = \sum_{t,t'}^T b_\alpha^{(2T-t-t')} s_{x_t x_{t'}}^\alpha
\tag{39}
$$

The separable equation is then $\frac{\partial u}{\partial t} = 2u \left[ \beta_{yx} - u \beta_{xx} \right]$, which in its partial fraction decomposed form is

$$
t = \frac{1}{2\beta_{yx}} \int_{u_0}^{u_f} \frac{du}{u} - \frac{1}{2\beta_{yx}} \int_{u_0}^{u_f} \frac{du}{u - \beta_{yx}/\beta_{xx}}
\tag{40}
$$

Integration of Eq. 40 yields

$$t = \frac{1}{2\beta_{yx}} \log |u|_{u_0}^{u_f} - \frac{1}{2\beta_{yx}} \log |u - \beta_{yx}/\beta_{xx}|_{u_0}^{u_f} \tag{41}$$

which upon reorganization gives the solution for training dynamics $u(\tau) = a(\tau)c(\tau)$

$$a(\tau)c(\tau) = \frac{1}{\left[\frac{1}{(a(0)c(0))} - \frac{\beta_{xx}}{\beta_{yx}}\right] e^{-2\tau\beta_{yx}/\gamma} + \frac{\beta_{xx}}{\beta_{yx}}}, \tag{42}$$

where $\gamma$ is the inverse of the learning rate. Eq. 42 is for a single mode $\alpha$, and there will be equivalent expressions for each of the $\alpha = \{1, 2, .. \min[N_x, N_y]\}$ modes.

In summary, we provided an expression for the learning dynamics of input and output modes in linear RNNs that encompass tasks with temporally correlated inputs. The primary assumptions that limit our current approach are 1) requiring input variability to be aligned with input-to-target mappings, 2) assuming normal recurrent dynamics, and most importantly 3) requiring that initial conditions of task parameters are in an orthogonal space with respect to the task geometry to facilitate factorized training with individual network modes.

## A.2 EXPANSION OF CL TRAINING TIME IMPROVEMENT

Here we add in a few intermediate steps to show the our core results of the conditions for training time improvement. Starting from the general condition:

$$t_{i\to 2} > t_{i\to 1} + t_{1\to 2}. \tag{43}$$

we first expand the the optimal solutions and initial conditions in terms of the recurrence-weighted singular values for all except solo task 1 and task 2 training,:

$$\frac{\gamma}{2\beta_{yx}^{(2)}} \left( \log \left| \frac{u^{*(2)}}{u_0} - 1 \right| - \log \left| \frac{\epsilon^{(2)}}{1 - \epsilon^{(2)}} \right| \right) >$$
$$\frac{\gamma}{2\beta_{yx}^{(1)}} \left( \log \left[ \frac{u^{*(1)}}{u_0} - 1 \right] - \log \left| \frac{\epsilon^{(1)}}{1 - \epsilon^{(1)}} \right| \right) +$$
$$\frac{\gamma}{2\beta_{yx}^{(2)}} \left( \log \left| \frac{\beta_{yx}^{(2)}/\beta_{xx}^{(2)}}{(1 - \epsilon^{(1)})\beta_{yx}^{(1)}/\beta_{xx}^{(1)}} - 1 \right| - \log \left| \frac{\epsilon^{(2)}}{1 - \epsilon^{(2)}} \right| \right) \tag{44}$$

Removing the common term (learning rate, accuracy term for task 2), and bringing all terms to both sides gives

$$\frac{1}{\beta_{yx}^{(2)}} \left( \log \left| \frac{u^{*(2)}}{u_0} - 1 \right| \right) - \frac{1}{\beta_{yx}^{(1)}} \left( \log \left[ \frac{u^{*(1)}}{u_0} - 1 \right] - \log \left| \frac{\epsilon^{(1)}}{1 - \epsilon^{(1)}} \right| \right)$$
$$- \frac{1}{\beta_{yx}^{(2)}} \left( \log \left| \frac{\beta_{yx}^{(2)}/\beta_{xx}^{(2)}}{(1 - \epsilon^{(1)})\beta_{yx}^{(1)}/\beta_{xx}^{(1)}} - 1 \right| \right) > 0 \tag{45}$$

Finally, multiplying everything by $\beta_{yx}^{(2)}$ and combining the logs of the middle term gives our final expression

$$\log \left| \frac{u^{*(2)}}{u_0} - 1 \right| + \frac{\beta_{yx}^{(2)}}{\beta_{yx}^{(1)}} \log \left| \left( \frac{\epsilon^{(1)}}{1 - \epsilon^{(1)}} \right) \left( \frac{u_0}{u^{*(1)} - u_0} \right) \right| - \log \left| \frac{1}{(1 - \epsilon^{(1)})} \frac{\beta_{yx}^{(2)} \beta_{xx}^{(1)}}{\beta_{yx}^{(1)} \beta_{xx}^{(2)}} - 1 \right| > 0 \tag{46}$$

## A.3 SIMULATION AND TRAINING METHODS

All simulations were performed in Python (3.11.6, torch 2.0.1). Numerical simulations were performed using gradient descent using a learning rate of 0.001 unless otherwise noted. Networks were

custom linear RNNs with 128 units and a single input and output channel. All initial conditions were initialized into a single input-output mode as specified by eq. 32 with the value $a = c = 0.01$. In practice, this meant setting a single entry $W_x[0, 1]$ and $W_y[0, 1]$ to 0.01, with the rest of the values initialized to zero. The recurrent weights were set to a diagonal matrix where all eigenvalues were $b = 0.96$.

For linear RNN simulations fitting that compared learning trajectories to theory, we first calculated the theoretical optimal solution using eq. 11, then trained the network until it converged to weights in an $\epsilon$ window of the optimal value. To calculate this optimal value, and to produce the theoretical wait time predictions and learning trajectory curves, we used analytically defined covariance matrices based on the defined task structure.

For nonlinear RNNs, we approximated the optimal solution by training the RNN until convergence, and then calculated the numerical training time in each task by retraining the network from scratch until it reached an $\epsilon$ window of the optimal solution.

When generating phase portraits and results for nonlinear RNNs across a range of task $\mathcal{T}_1$ accuracies, we only modulated one task parameter at a time, and kept all parameters across $\mathcal{T}_1$ and $\mathcal{T}_2$ equal. When modulating covariance strength $\boldsymbol{S}_{x_t y}$, we held $\boldsymbol{S}_{x_t x_{t'}} = 1.0$. Similarly, when modulating $\boldsymbol{S}_{x_t x_{t'}}$, we held $\boldsymbol{S}_{x_t y}$=1.0 In sec. 3.2, when modulating $K$ we held $\kappa$=0.5; when modulating $\kappa$, we held $K = 0.5$.

### A.4 SUPPLEMENTAL ANALYSES

#### A.4.1 NUMERICAL COMPARISONS

Here we show numerical simulation fits to theory for each parameter regime studied in the main work, as well as the corresponding optimization trajectory for ReLU Rnns.

#### A.4.2 MISMATCHED ACROSS-TASK GEOMETRY

In order to study an extension beyond our theory for cases in which the factorized modes of task 1 no longer align with task covariances in task 2, we simulated a system with two inputs and two outputs, which gives two factorized modes in the network. We focused on examining if relative changes in input-target covariance strength would be recapitulated even in this scenario, by seeing if training times reflected the results Fig. 2A-C. The target task $\mathcal{T}_2$ here had $\boldsymbol{S}_{x_t y} = \text{diag}[1.2, 1.0]$ for its input-target covariance singular values, and $\boldsymbol{S}_{x_t x_{t'}} = \text{diag}[1.2, 1.0]$ for its input-input covariances singular values. To avoid any degeneracies in the network we set the two related recurrence eigenvalues to $\lambda = [0.96, 0.94]$, and set the remaining values to 0. The desired accuracy for both tasks was set to 95%. $\boldsymbol{S}_{x_t y}$ for task 1 always kept one mode fixed to the same value for $\mathcal{T}_2$, but we set its second singular value to be either larger or smaller.

The left eigenvectors for $\boldsymbol{\Sigma}_{x_t y}$, $U_{xy}$ (eq. 25), were set to the identity. Importantly, we looked at training times in the "geometry-matched" regime where $U_{xy}$ was also the identity, as well as a "geometry-mismatched" regime in which $U_{xy}$ was rotated with a 2D rotation matrix by $\pi/4$. We found that the qualitative trend seen in Fig. 2A-C holds, which is that relatively stronger $\mathcal{T}_1$ $\boldsymbol{S}_{x_t t}$ strength led to faster training, and vice versa. Moreover, we found that there was some representational rotational effects that occurred for $\mathcal{T}_1$ having weaker input-target covariance strength, which we measured by calculating the normalized inner product between the input matrix $W_x$ over training to its final solution at the end of $\mathcal{T}_2$.

#### A.4.3 CL EFFECTS WITHOUT RECURRENCE

To make a comparison to feed-forward networks, we also simulated the theoretical results of varying covariance strength for the input-target covariance when the recurrent weights were set equal to the identity matrix. As mentioned in the main text, in this special case of constant singular values and perfectly stable dynamics ($b = 1$), this recapitulates the theoretical results of feed-forward networks studied in Saxe et al. (2014). Here we demonstrate that the same trends in CL improvement exist in this setting, though the magnitude of the effect is reduced.

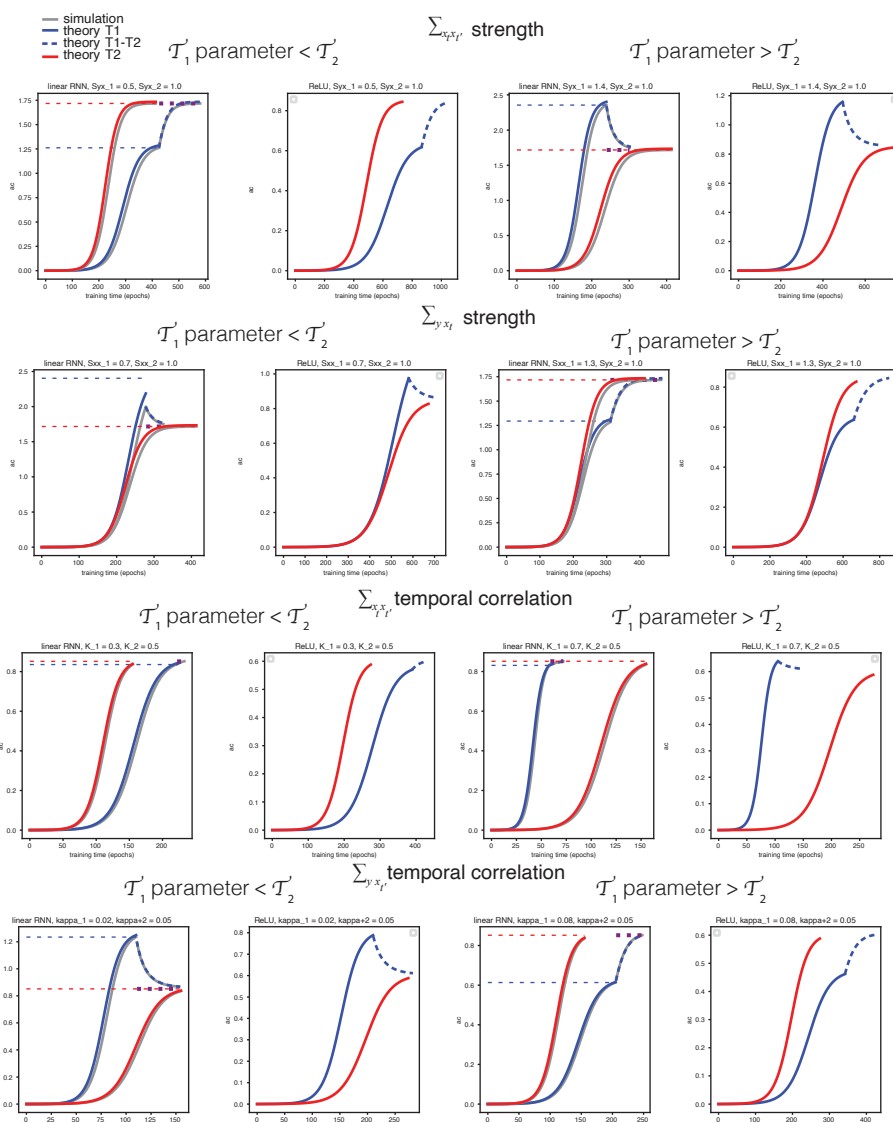

Figure 5: Numerical fits of optimizations using Curriculum learning (blue) vs. direct training of the target task (red). ReLU learning trajectories are also shown. Each row corresponds to manipulating one aspect of task covariance structure, and here we provide examples for both a parameter setting in task 1 that is smaller than task 2 (left 2 plots) as well as larger (right two plots). Theoreetical optima are shown for each task with horizontal dotted lines

### A.4.4 JOINT TRAINING WITH RECURRENCE

We also sought to understand if our qualitative results about CL effectiveness could hold when jointly training recurrent weights alongside input and output weights. Introducing training of recurrent weights is more numerically unstable then just input and output weight training because of exploding/vanishing gradient issues; in order to make a tractable comparison we focused on initial conditions near an optimum that we had originally found when training the inputs and outputs, then saw how the parameters changed from there. Specifically, we looked at scenarios analogous to Figure 2A-C, where we modulated the strength of input-target covariance strength. We trained systems with 100 timesteps with a much smaller learning rate ($\gamma = 10^{-7}$), and we chose initial conditions where the single nonzero eigenvalue of the recurrent weight matrix was set to $b = 0.96$, which was

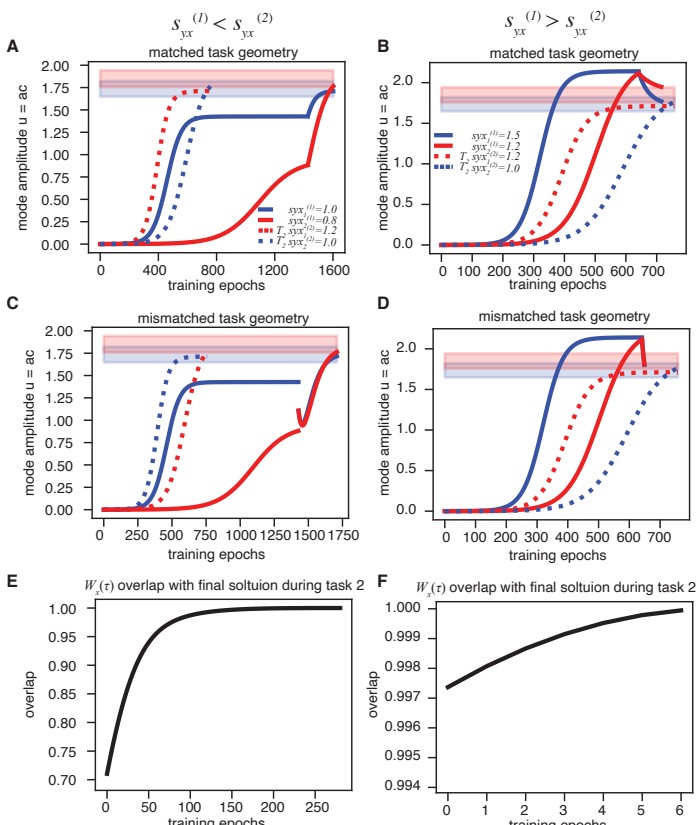

Figure 6: Effects of across-task, mismatched geometry. Training time in a 2D system with varying input-target covariances was studied when covariance eigenvalues for $\mathcal{T}_1$ and $\mathcal{T}_2$ were aligned, or mismatched. Left column is for the case where one input-target singular value is smaller in $\mathcal{T}_1$ than in $\mathcal{T}2$, and vice versa in the right column. **A-B** Training for the CL sequence vs. direct training when task geometry is matched. Blue lines correspond to the mode that has singular value strength that is matched across the tasks, and red lines denote the one that is different. Dotted lines denote direct training of $\mathcal{T}_2$, solid lines denote the CL sequence training. **C-D** Training for the CL sequence vs. direct training when task geometry is mis-matched. **D-E** Overlap of the input weights throughout training $W_x(\tau)$ with the final solution for input weights $W_x(\tau_f)$. Shaded regions in panels **A-D** denote the one-sided 95% accuracy level around the theoretical solution for each factorized mode.

the value used in the rest of our work. We then scaled down the optimal input and output values by 50%, then trained all parameters to a loss $\mathcal{L} < 10^{-5}$.

We found that the inputs and outputs barely changed in magnitude, and that the recurrent weight changes dominated the optimization in scenarios where task 1 had a weaker input-target covariance (Fig. 8A), or a stronger one (Fig. 8B). We did find that the qualitative trend where stronger task 1 covariance strength coincided with faster training due to curriculum learning, and longer training coincided with a weaker covariance in task 1.

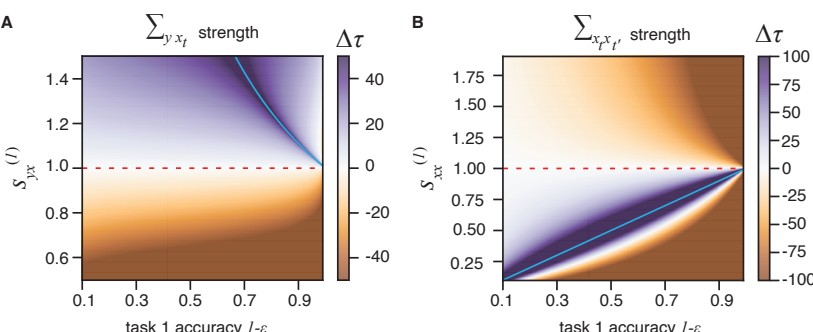

Figure 7: Theoretical predictions for varying input-target covariance strength with perfect recurrence (b=1). Results are as in Fig. 2A,2D.

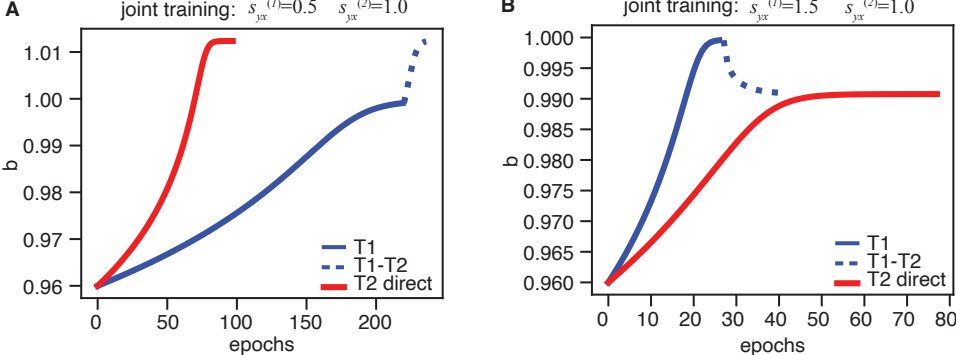

Figure 8: Recurrent weight change over joint training of all network parameters. Input and output weights were nearly constant for the entire optimization, and are not shown. **A** Training where $\mathcal{T}_1$ had a weaker input-target covariance strength than $\mathcal{T}_2$. **B** Training where $\mathcal{T}_1$ had a stronger input-target covariance strength than $\mathcal{T}_2$. Red lines denote direct task 2 training, and blue lines denote CL sequence training. Blue dotted line shows where $\mathcal{T}_2$ training begins.

