# OpenReview forum: "Theoretical foundations of curriculum learning in linear RNNs"
_ICLR.cc/2026/Conference — Submitted to ICLR 2026_

### Official Review · Reviewer_wg5z · 2025-10-27

**Soundness:** 3
**Presentation:** 3
**Contribution:** 1
**Rating:** 2
**Confidence:** 3

**Summary:**

This paper studies the training time of linear RNNs in two scenarios: (i) learning a single "target" task; (ii) first learning an "easier" task and then learning the "target" task when initialized from the first task. By simplifying the problem via an alignment assumption between tasks, the authors derive closed-form solutions for the time required for both scenarios. These times depend on the task via the covariances of the data. They analyze when (ii) is faster than (i) by considering training the first task to an $\epsilon$ error. They validate their analysis in linear RNNs and provide further evidence from nonlinear RNNs.

**Strengths:**

The paper has a refreshing style based on training dynamics analysis for curriculum learning. It provides sharp and testable predictions. As far as I'm aware, this is novel. The theory is also presented clearly.

**Weaknesses:**

1. The main issue with the paper is the assumption that the training time of a gradient flow dynamics quantifies the hardness of the task. The speed of learning is sensitive to the learning rate. Based on the authors' definition, one can adjust the learning rate to change the hardness of the problem, which does not make sense. This is very much reflected in the results of the paper. Using stronger singular values in the first task is a way of increasing the learning rate.

2. Related to the point above, the hardness of a task should incorporate something about the statistical hardness of the task. It is not at all clear that the notion the authors proposed aligns with the statistical difficulties of problems. For example, consider a regression problem where the data is scaled by some constant. This is the same problem and the difficulty should be the same. Therefore, the results need to take into account some form of normalization, which is beyond the learning rate discussion. Ideal results with training dynamics would look like this: the first task allows an efficient recovery of the domain of the target task and then the second task is learned much more efficiently.

**Questions:**

1. Can you comment on the example I have given regarding scaled regression? Is it true that your model considers this as a simpler task? Can't you derive the benefits of curriculum with simple learning rate scheduling then?

2. What is the difference between linear networks and linear RNNs when we just focus on prediction at the last token? I see that we get more complicated covariances but aren't they the same up to some transformations in the data?

---

> ### Author Response · Authors · 2025-11-21
> **Response to Reviewer Weaknesses and Questions**
>
> Weakness issues are recapitulated by the specific questions asked by the reviewer, so we answer The Questions here:
>
> Q1: First we want to note that relative training benefits between two tasks are learning-rate invariant: While the absolute training time of a task depends upon choice of learning rate (eq. 12, where gamma is the inverse of learning rate), the expression for studying relative training time speedup  (Eqs. 15-16) cancels that factor out.
>
> Importantly, the rescaling of units in regression is not the right intuition for what is going on here.  The units of signal are not something that can be independently manipulated but they define what the task is. More generally, measures of task difficulty are hard to formalize, so we went for the pragmatic approach: a task is harder if it takes more effort (more training epochs) to obtain.
>
> Q2: The question could be interpreted in two ways which we address both here.
> First, how would our task setup and results differ if the loss depended on outputs at other times in the trial, not just the end? The input-target covariance would become more complicated, including an extra time dimension that would transform it from a 3-D tensor (input dim X target dim X input time dim) into a 4-D tensor (input dim X target dim X input time dim x target time dim ). The learning dynamics of this setup was documented and studied in Proca et. al., ICML 2025, and in the context of our work here it would extend our analysis of CL effectiveness to include time dependence between the inputs and the time dependence of the targets. Since we have already used the temporal dependence among inputs to build intuitions about how this aspect of a task can impact CL, we have decided not to study this use case here.
> Second, there is a question about whether or not a coordinate transformation can reduce complexity in task covariances. One cannot generally remove the task covariance complexity of multiple tasks simultaneously with a single coordinate transformation. This is why we chose to study the task covariances in their fullest form, without coordinate transforms.

---

> > ### Comment · Reviewer_wg5z · 2025-11-26
> >
> > > Q1: ...
> >
> > This is true only if one chooses the same learning rate in all phases ($t_{i\to1}, t_{i\to2}, t_{1\to2}$). I don't see why this is a reasonable assumption. The learning rate depends on the scale of the target function in regression problems. Overall, the learning dynamics and statistics of $f_\star(\cdot) = \langle \cdot, w_\star \rangle$ are exactly the same as $c f_\star$ where $c$ is some constant. Isn't it reasonable that the learning rate $\gamma$ depends on $c$? I agree that the task difficulty is hard to formalize but I think your way of formalizing it doesn't even captures the proxy "effort".
> >
> > > Q2: ...
> >
> > I am asking if your analysis, with all the simplifying assumptions on the network and the loss only on the last token, can be seen as an analysis of an equivalent problem where the model is a linear network and data has an interesting covariance structure, possibly of higher dimension without any temporality. My intuition is that $W_h$ being frozen should give us some sort of equivalence among these problems?

---

> > > ### Author Response · Authors · 2025-12-01
> > > **response to follow-up comments**
> > >
> > > SUMMARY:, we feel that the reviewer’s low Contribution score does not hold when the intuition behind their argument is held up to additional scrutiny. Our theory attempts to tackle learning problems in their richest possible form so that the constraints of a broader class of learning systems (i.e., biological ones) can be studied. While scores can no longer be adjusted because of the data leak, we hope that the area chair sees that we have adequately addressed the reviewer's concerns, and given evidence that our overall score should have been higher.
> > >
> > > Q1: The details of the regression example seem to obscure a broader critique, namely that problems from the same task class possess the same level of intrinsic difficulty, and that –under some appropriate transformation– all such tasks become equivalent (e.g., all regression problems are equally difficult, modulo data whitening and target re-scaling). This issue appears to be what drove the reviewer’s low “Contribution” score, as it would dilute our work’s impact on solving difficult tasks using pretraining.
> > >
> > > We argue that such standardization is not always possible or useful, especially when thought about not in abstract mathematical terms but w.r.t. multitask goals that encompass the constraints in biological learning, a much broader scope than pure machine learning.
> > >
> > > A  good counter-example for the rescaling of regression type of argument is prediction: one step ahead prediction is easier than long horizon forecasting, as errors accumulate over time. Forecasting horizon is thus the intuitive dimension of task difficulty. One could re-define the units of time to make any task a one step ahead prediction but that would still require at the very minimum appropriately rescaling the noise, making different horizon problems not completely equivalent. Importantly, in biological circuits or RNN models of them, the natural time constants are not something that can be changed arbitrarily from task to task. This means that despite some abstract mathematical equivalence such tasks are not equivalent problems for the learning system. Arguably, temporal integration of the kind considered in our analysis falls in this same category.
> > >
> > > Second, although adjusting learning rates to the task is indeed practically useful to speed up learning in single tasks, this is not a general approach to learning tasks in sequence. Dynamic learning rates would be hard to determine in a multi-task continual learning setup, especially if tasks are interleaved randomly, e.g. to encourage active learning (Carvalho, Goldstone, 2015). Assuming a fixed learning rate and analyzing relative speeds of training in units of that seems sensible both for practical considerations and in terms of relevance for biology.
> > >
> > > Q2:  We thank the reviewer for clarifying the question. The specific assumptions used in this work’s formulation (linearity, frozen recurrence, final timestep loss function) may indeed allow for further problem simplification that reduces the temporal structure at least at the level of single tasks;  the usual caveats about putting multiple tasks in the same representational frame apply. Our approach was to study simple cases as a way to build intuition that can be ported into situations where the theory may not formally hold, but the insights do. In that vein, we do not look to reduce our setup to a minimal version, but try to think of them as instances of a broader problem class; numerically, we aim to study cases with the richest task structure that the theory can accommodate.

---

> ### Author Response · Authors · 2025-12-03
> **Summary for Area Chair**
>
> This reviewer felt that our work was sound and well-presented in a “refreshing style,” but had a main issue with the work's contribution to the field. They felt that the premise for how we studied the tasks (i.e., task hardness is given by training time) was fundamentally not justified, and undercuts the necessity of the work. They used a specific example of linear regression to present their point, and asked follow-up question about this specific use case. We responded to the reviewer about how these specific points they brought up are not valid in the context of this work. They replied with additional specific questions about this use case. We then realized that the reviewer was presenting this use case to hit at a larger question, which was that the type of task should dictate its hardness, and that trivial transformations within a class of tasks can be performed to standardize them to the same level of hardness. We addressed this larger question with multiple counter-examples where the logic of standardizing tasks in the way does not hold. This demonstrated that their primary concern about our work was not well supported, and that our score should have been revised. Because of the data breach, the reviewer was unable to respond to our final comment, but we believe that it provided substantial counter-evidence to their original claim, and that our score would have been improved by this fruitful discussion with the reviewer.

---

### Official Review · Reviewer_c6Wu · 2025-10-31

**Soundness:** 4
**Presentation:** 3
**Contribution:** 2
**Rating:** 6
**Confidence:** 4

**Summary:**

This paper studies the effect of curriculum in a linear RNN setting with fixed recurrent weights and trainable input-to-hidden and hidden-to-output weights. It identifies task statistics that yield faster learning from a curriculum as opposed to direct training. The results are verified in simulation and qualitatively hold in ReLU RNNs.

**Strengths:**

The paper presents new exact solutions to the learning dynamics of a linear RNN with fixed recurrent weights.

The results identify the key dataset properties in this setting which enable curriculum learning to outpace direct training.

The paper is clear and the figures are to a high standard.

**Weaknesses:**

The studied model is a particularly simple form of RNN in which the recurrent weights are not trainable. The paper could be strengthened by investigating whether qualitatively similar results hold when recurrent weights are trained as well.

The tasks studied are versions of learning to integrate an input. While this is an interesting task, it would be useful to understand the limits of the theory for other types of tasks.

The paper could benefit from discussing other theoretical work on curriculum, for instance the work of Stefano Sarao Mannelli.

**Questions:**

Does a qualitatively similar picture hold when recurrent weights are trained?

Does a feedforward network yield similar optimal curricula or does recurrence make a different set of curricula beneficial?

---

> ### Author Response · Authors · 2025-11-21
> **Responses to Weaknesses and Questions**
>
> Weakness 1 is recapitulated by Question 1. Weakness issues 2,3 are addressed here, followed by the Questions:
>
> W2: While we agree that a broader class of tasks would increase the impact of the work, linear RNNs cannot do anything other than spatiotemporal filtering, so integration is a good representation of the tasks that these networks can possibly perform. To extend the scope beyond our linear theory, we are planning to use numerics to a nonlinear variant where a binary decision gets computed at the end of a trial, an analog of simple decision making. In practice, this means using a binary cross entropy loss instead of mse. We are in the process of simulating this decision-making task in the same regimes for input-target covariance strength in task 1 being stronger or weaker than task 2. We will add it to the appendix once complete.
>
> W3: The reach of CL in machine learning is very broad, and we aimed to be honest about the scope of our theory’s impact, and initially decided to focus on what progress had been made on this front. We agree with the reviewer that a more comprehensive discussion of the theory of CL would benefit the paper.
> We could not identify a relevant first-authored paper by the author Stefano Sarao Mannelli on the topic, and we assume that the reviewer is referring to one of two papers: The theory of CL work by Saglietti, Mannelli, and Saxe (Neurips 2022)  pertains to using curriculum learning in feed-forward networks with teacher-student networks. While this training setup is substantially different from our recurrent linear RNN setup, i.e. no teacher-student networks, it does provide closed form expressions for the performance accuracy in CL sequences. This is in the spirit of our overall aim, which is to provide analytical results for CL performance, and we will include it in our discussion of relevant work. The theory of compositional pretraining by Lee, Mannelli, and Saxe (ICML 2024) studies feed-forward networks in a reinforcement learning setting, and derives training time expressions for different compositional curricula. We have included this work in our introduction as well.
>
> Q1.We are finalizing theoretical analysis for the joint optimization of recurrent and input/output modes in a tractable scenario that was described in (Schuessler et. al, 2020). At the moment, we are validating that part of the theory. Briefly, our theory utilizes the fact that network states can be more easily calculated when they converge to a fixed point. This limit can lead to a 2D system of nonlinear ODEs for parameter updates, and suggests an approximation where a collective variable can describe optimization of all parameters in closed form. We have included numerical results showing how CL intuitions from input/output weight training still hold in this setup: We jointly optimized all parameters in scenarios with varying input-output covariance strength (analogous to Fig. 2 B-C). Recurrent weight updates dominated the optimization, and the qualitative trends in the input/output training case still hold (included in Appendix).
>
> Q2: Feedforward networks were studied comprehensively in Saxe 2014, which derived a closed-form expression for training dynamics, as well as optimal solutions, but had no reference to CL and used no covariance across inputs. We haven’t investigated feedforward nets extensively, but note that in the limit of recurrence eigenvalue b = 1, the RNN functions effectively as a feedforward network so the qualitative trends will persist. We have included a numerical demonstration of this in the Appendix, and mentioned the correspondence in the main text.

---

> > ### Author Response · Authors · 2025-12-03
> > **Follow-up to Weakness 2**
> >
> > Unfortunately we were unable to complete the analysis of the decision-making variant of the task in the rebuttal time frame. It is an interesting question, and will be an aspect of our future work on this project. We thank the reviewer for the suggestion.

---

> ### Author Response · Authors · 2025-12-03
> **Summary for Area Chair**
>
> This reviewer also found our work to be very sound, relevant to the ICLR audience, but they felt that it had a slightly diminished contribution compared to the first reviewer. They asked several questions similar to the first reviewer about extending our results beyond the scope of the theory’s main analysis, as well as providing more context for how this work fits into the broader literature.  We addressed nearly all of these concerns with follow-up simulations, revisions to our manuscript, and comprehensive replies to the reviewer. While we were unable to complete a reviewer request to show analysis on a different class of tasks in the rebuttal time frame,  we still feel that the reviewer comments improved the work, and paved the way for additional research questions.

---

### Official Review · Reviewer_UJEY · 2025-11-02

**Soundness:** 3
**Presentation:** 3
**Contribution:** 3
**Rating:** 6
**Confidence:** 4

**Summary:**

The paper develops a theoretical account of curriculum learning (CL) in linear RNNs and validates key predictions in simulations (with a small nonlinear check). Using recent analyses of learning trajectories in linear RNNs (Proca et al., 2025), the authors study training two related tasks in sequence and ask when pretraining on task T1 accelerates learning of target task T2. The main results are: (i) sequencing by task covariance matters—pretraining that increases input–target covariance strength and scale can provably reduce training time for T2; (ii) counterintuitively, stopping T1 early (sub-optimal accuracy) can yield faster T2 learning; and (iii) the theory gives explicit time-to-convergence formulas and phase-plane predictions, corroborated numerically.

**Strengths:**

1. The abstract and discussion cleanly articulate what structural aspects of the two tasks (covariance strength/temporal structure) drive speedups, with the “stop T1 early” prediction highlighted as surprising.

2. The appendix lays out the model and derivation path (extending recent work), including the linear RNN equations and loss setup.

3. The paper motivates CL/pretraining’s mixed empirical record and positions the analysis within that context.

4. Simulations, including a nonlinear-RNN sanity check (Fig. 4), exhibit the same qualitative dependencies predicted by the theory across different task-covariance regimes.

**Weaknesses:**

1. Results are derived for pairs of “similar” tasks with shared geometry, and training focuses on input/output weights with predefined recurrence—limiting generality.

2. The main emphasis is learning speed; robustness/generalization/noise sensitivity are left to future work.

3. The nonlinear experiments are positioned as qualitative trend checks rather than tight quantitative tests of the theory.

4. The paper does not cite some relevant work, eg papers that analyzed curriculum sequencing, representational transfer, and gradient-alignment mechanisms in RNNs, like Kepple, Engelken, Rajan, ICLR, among others. Those works anticipated aspects of the current analysis—particularly the role of inter-task geometry in shaping convergence—so their absence weakens contextualization. The related-work section should explicitly discuss how this study’s theoretical framework extends or differs from that earlier line of curriculum-learning theory.

**Questions:**

1. Can you add quantitative error bars comparing predicted vs. observed time-to-convergence in nonlinear RNNs across the covariance sweeps in Fig. 4?

2. What breaks first if T1 and T2 do not share task geometry (e.g., rotations/compositional changes)? Any preliminary results on the “rotation to factorized regime” you outline?

3. How would updating Wh alter the phase-plane and convergence-time analysis? You note it as feasible for long-timescale computations—any tractable subcase?

4. Given the “stop early” result, could you propose a curriculum-selection heuristic (e.g., proxy measures of input–target covariance strength/scale) and a stopping rule for T1? (Pointers exist but a recipe would help.)

5. Any insight on whether the same covariance principles predict robustness/generalization improvements, not just speed?

---

> ### Author Response · Authors · 2025-11-21
> **Response to reviewer Weaknesses and Questions 1,2,3,5**
>
> Weaknesses 1-3 are addressed by the questions section. We address Weakness 4 first, then address Q 1,2,3 and 5. Q4 (stopping heuristic) is an important question that we address separately in another reply.
>
> W4: The reach of CL in machine learning is very broad, and we aimed to be honest about the scope of our theory’s impact, and initially decided to focus on what progress had been made on this front. We agree with the reviewer that a more comprehensive discussion of the theory of CL would benefit the paper.  Reviewer 1 claims that Kepple et. al.  anticipates aspects of the role of inter-task geometry in shaping covariance. In our reading, this work utilized different CL sequences to uncover learning principles in networks that had nearly indistinguishable performance, but may have learned using different learning rules. We have included this work in the introduction because it speaks to the novel ways that CL can be used, but we would require a bit more explanation from the reviewer as to how it speaks to a theoretical knowledge of CL. Specifically, i) it used heuristic curriculum learning strategies borrowed from experimental work, ii) it provided no first-principles assessment of why the CL strategies succeeded or failed, and iii) there appears to be no concrete mention of the relationship between across-task geometry and CL effectiveness (note: there is not mention of “geometry” in this manuscript).
>
> Q1. Our theory holds only for linear systems, so unfortunately we cannot compare the empirical optimization speed to a theoretical standard. There was also no noise in our networks over which to gather statistics. We aimed to show that extensions beyond the theory preserve the qualitative insights in a linear system, to show theory’s broader impact. We have emphasized that this analysis is an extension beyond the scope of the theory in the main text.
>
> Q2. We simulated the optimization of a second task that does not share the same geometry as task 1, and investigated the case when input/output weights rotate between the two reference frames set by the task covariances – in a 2D input and 2D output setup in which task 2 has diagonal task covariances, but Task 1 has rotated task covariances. In the more difficult regime, this takes longer to optimize, and when we examined the parameter overlap with the final task 2 solution in that case, we found that the system did have a rotation in its input weight representation (included in appendix).
>
> Q3. We are finalizing theoretical analysis for the joint optimization of recurrent and input/output modes in a tractable scenario that was described in (Schuessler et. al, 2020).  At the moment, we are validating that part of the theory. Briefly, our theory utilizes the fact that network states can be more easily calculated when they converge to a fixed point. This limit can lead to a 2D system of nonlinear ODEs for parameter updates, and suggests an approximation where a collective variable can describe optimization of all parameters in closed form.  We have included numerical results showing how CL intuitions from input/output weight training still hold in this setup: We jointly optimized all parameters in scenarios with varying input-output covariance strength (analogous to Fig. 2 B-C).  Recurrent weight updates dominated the optimization, and the qualitative trends in the input/output training case still hold (included in Appendix).
>
> Q5. We iterate several caveats to our analysis that we ought to have highlighted better in our work, which will hopefully explain why training speed is the most crucial aspect of CL to study here. In general, the analyses needed might not be tractable in the rebuttal timeframe, and are best suited for future work where their impact can be more comprehensively studied.
>
> "Generalization improvements": There is growing evidence that connects empirical results in behavioral shaping as potentially useful in machine learning problems (Lee et al. 2024; Hocker et al. 2025), and that such pretraining can lead to more stereotyped solutions. This is important for CL when network tasks are complex, with many potential solutions. In our work, the integration tasks are convex optimization problems with unique optimal solutions u*. u* is unique for a given factorized mode, which is a product of contributions from input and output weights (u* = a*c*). While formally there is a degenerate set of optimal solutions that could solve this task by shifting contributions from a or c, we decided not to pursue this aspect of generalization here as it is somewhat trivial.
>
> "Robustness" could be defined in multiple ways.  To address robustness to input noise, we would derive the expected change in training loss with respect to input noise. Robustness to noise in the training parameters would require defining second-order sensitivity of the loss with respect to the training parameters (i.e, the Hessian of L), which is a considerably more complicated analysis.

---

> > ### Author Response · Authors · 2025-12-03
> > **Follow-up response to Q3**
> >
> > After performing additional analysis on the joint optimizations of recurrent and input/output parameters, we found that our theoretical work requires assumptions that are not regularly satisfied when performing numerical optimizations. Specifically, the simplified form that the network takes in the long time limit assumes that recurrent weights will not be tuned near 1, as this will lead to divergent network states for constant, non-zero inputs. This is an assumption that is often broken in practice. We have kept the numerical simulations in our supplemental section since they demonstrate that the qualitative trends found in our theory appear to hold, and refer to it in the main text. We have decided not to include the theoretical analysis in this work, as it will require effort beyond the scope of this submission to identify methods to satisfy its key assumptions.

---

> ### Author Response · Authors · 2025-11-21
> **Response to Q4**
>
> Q4. This is an excellent question. Speed is related to input-output cross correlations, which can be measured empirically. We propose a heuristic for stopping task 1 based on the relative difference in task covariances, based primarily on the location of training time singularities, which we sketch out below. Our general approach is the following: i) locate a theoretical feature that enforces faster learning in our theory, ii) build evidence for this feature’s importance through empirical evidence, and iii) identify useful and tangible signatures of its presense/absence that can be utilized in more practical scenarios as stopping criteria.
>
> i) In eq. 17 and the final term in eq. 16, we identify the conditions in which a singularity in the difference in training time between directly training task 2, and using a CL approach would occur. This corresponds to the case where the task 1 solution (to a desired accuracy) directly corresponds to the task 2 solution. A desired heuristic would ask “are there ways to find an initial task and accuracy that land me near this singularity?”. Additionally, this expression gives the functional form for the nonlinear relationship between task 1 accuracy, and relative task structure.
>
> ii) in practice, locating this singularity in a real-world task would be quite difficult, but our empirical results in Figs. 2 and 3 demonstrate that the effect of this singularity appears to provide the support for faster training times across a large range of accuracies. They also demonstrate an important guiding principle for the trends between task 1 accuracy and relative task structure: For tasks with larger differences in covariance strengths (Fig 2A,C) the difference, lower task 1 accuracies are beneficial. Similarly, for larger timescale differences between the tasks (Fig 3D), the sooner one should stop training on the first task. This is one guiding principle that comes from our theory that we believe can be used in practice in complex tasks.
>
> iii) A more pragmatic question is how to ground the ideas in points i) and ii) into a practical test during training. We are still working on this, but our thought is to look at how changes in training losses and parameter gradients might hint that you are near this singularity regime.

---

> > ### Author Response · Authors · 2025-12-03
> > **Follow-up response to Q4**
> >
> > We wanted to update the reviewer and area chair about how we ultimately incorporated their suggestion about the stopping heuristic. We have incorporated our general approach about designing a heuristic into the discussion section of the work, as well as the practical advice for a stopping heuristic for this task: Since the functional form of parameter training dynamics is sigmoidal, there is a straightforward stopping heuristic for this class of tasks:  While training an initial task, monitor the loss of a target tasks for a sharp decrease in the loss, followed by saturation. This “rise then plateau” behavior of the second task is the key signature for when to stop optimizing task 1, and then fine tune the parameters to perform task 2, as it is now already near its optimal value. We also mention in the discussion section that the nature of the stopping heuristic will be task specific, as not all losses will have sigmoidal training dynamics that can take advantage of this approach. This was a great question posed by the reviewer that will be the focus of our follow-up work on this manuscript.

---

> ### Author Response · Authors · 2025-12-03
> **Summary for Area Chair**
>
> This reviewer found our work to be sound, relevant to the ICLR audience, and that it provided novel, “surprising” results about how to perform curriculum learning. They had additional questions about extending our theory’s results beyond the scope of the main analysis, providing more context for how this work fits into the broader literature, and if there were practical heuristics for how to implement the insights from our work in a practical way. We addressed all of these concerns with follow-up simulations, revisions to our manuscript, and comprehensive replies to the reviewer. We thank the reviewer for their questions, and believe that it has made our submission substantially stronger.

---

### Meta-Review · Area_Chair_BdPe · 2025-12-24

**Summary:**

The paper analyzes the training dynamics of linear RNNs for curriculum learning. The paper establishes certain conditions under which training on one task can help in learning the second task.

The paper received mixed reviews. Reviewers UJEY and c6Wu appreciated the thoroughness of the theoretical analysis for the considered problem. At the same time, both these reviewers also expressed concerns regarding the strong assumptions under which the results are shown, and limited scope of the analysis (such as only linear RNNs with recurrent weights not trainable, considering integration tasks with shared eigenvectors). Reviewer wg5z had a more negative assessment, and expressed concerns regarding the fact that the result only establishes differences in the training time of a specific algorithm, i.e. gradient descent with a fixed learning rate, instead of more generally investigating the statistical hardness.

Overall, while the paper has merit, it is marginally below the acceptance bar. The assumptions are rather strong which make the setting a bit stylized.

**Reviewer Concerns:**

The major concerns of the reviewers still stand. Reviewers UJEY and c6Wu expressed concerns about limited scope. The authors mentioned some additional results they may have, perhaps these can make a future revision stronger.

Reviewer wg5z's comment also stands. It is not clear if the authors fully understood this concern. The reviewers remark is that the difference in training times is only for a specific algorithm---gradient descent with a fixed learning rate. The paper does not show the difficulty of learning the second task if a different algorithm is used, such as a different learning rate. This gets to the inherent statistical hardness of learning the second task after learning the first one. I think this is a valid concern, though one could argue that just understanding the training dynamics of gradient descent for a fixed learning rate is interesting as well.

**Reviewer Scores:**

It is possible that all reviewers keep their scope if they had been able to participate fully in the discussion.

---

### Decision · Program_Chairs · 2026-01-26

Reject